ViralPlaque: a Fiji macro for automated assessment of viral plaque statistics

Cacciabue Marco cacciabue.marco@inta.gob.ar
Currá Anabella
Gismondi Maria I. gismondi.maria@inta.gob.ar
1 Instituto de Agrobiotecnología y Biología Molecular (IABiMo), Instituto Nacional de Tecnología Agropecuaria (INTA), Consejo Nacional de Investigaciones Científicas y Técnicas (CONICET) , Hurlingham , Buenos Aires , Argentina
2 Departamento de Ciencias Básicas, Universidad Nacional de Luján , Luján , Buenos Aires , Argentina
Gomez Shawn
Electronic publication date: 2019 Sep 24
Publication date: 2019
Volume: 7
Electronic Location ID: e7729
Received 2019 Jun 24; Accepted 2019 Aug 22
Copyright: ©2019 Cacciabue et al.
Copyright year: 2019
Copyright holder: Cacciabue et al.
License: This is an open access article distributed under the terms of the Creative Commons Attribution License, which permits unrestricted use, distribution, reproduction and adaptation in any medium and for any purpose provided that it is properly attributed. For attribution, the original author(s), title, publication source (PeerJ) and either DOI or URL of the article must be cited.
License URL: https://creativecommons.org/licenses/by/4.0/

Keywords: ImageJ macro, FIJI, Machine learning, Lytic virus, Cytopathic effect, Plaque assay, Plaque size

Funding: Instituto Nacional de Tecnología Agropecuaria and Agencia Nacional de Promoción Científica y Tecnológica PICT 2014-982 PICT 2016-1327 PICT 2017-2581 This work was supported by Instituto Nacional de Tecnología Agropecuaria and Agencia Nacional de Promoción Científica y Tecnológica (PICT 2014-982, PICT 2016-1327 and PICT 2017-2581). The funders had no role in study design, data collection and analysis, decision to publish, or preparation of the manuscript.

==============================
Plaque assay has been used for a long time to determine infectious titers and characterize prokaryotic and eukaryotic viruses forming plaques. Indeed, plaque morphology and dimensions can provide information regarding the replication kinetics and the virulence of a particular virus. In this work, we present ViralPlaque, a fast, open-source and versatile ImageJ macro for the automated determination of viral plaque dimensions from digital images. Also, a machine learning plugin is integrated in the analysis algorithm for adaptation of ViralPlaque to the user’s needs and experimental conditions. A high correlation between manual and automated measurements of plaque dimensions was demonstrated. This macro will facilitate reliable and reproducible characterization of cytolytic viruses with an increased processing speed.

Introduction

Plaque assay is a typical test originally used for bacteriophage characterization and subsequently adapted to eukaryotic viruses to estimate infectious titers or to perform clonal purification of viral agents (D’Hérelle & Smith, 1926; Dulbecco, 1952). Beyond these uses, formation of viral plaques is a relevant phenotypic feature that can withhold essential information of the virus under study. Indeed, plaque statistics (size, clarity, border definition and distribution) can provide important information on the replication kinetics and virulence factors of a virus. Morphology of plaques has been reported to help distinguish viral isolates and is currently used as an indicator of virus attenuation (Tajima et al., 2010; Goh et al., 2016; Kato et al., 2017; Fan et al., 2018; Moser et al., 2018; Schade-Weskott, Van Schalkwyk & Koekemoer, 2018).

Basically, during a plaque assay, a monolayer of susceptible cells is exposed to a serially diluted lytic virus (commonly 5–100 virions per well). An immobilizing overlay (typically agarose or methyl cellulose) is used to prevent uncontrolled viral spreading through the liquid medium. Following initial infection, regions of dead cells are formed as viral replication cycles unfold, forming individual plaques (Baer & Kehn-Hall, 2014). Typically, cellular monolayers are then fixed and counterstained with neutral red or crystal violet solutions, which facilitates the identification of the plaques formed. After these steps, plaque statistics (e.g., plaque number or dimensions) can be obtained from the infected cellular monolayer by direct visualization and manual recording. Of note, plaque morphology can vary depending on growth conditions and between viral species (Baer & Kehn-Hall, 2014).

Although manual determination of plaque number can be highly sensitive (i.e., it is possible to detect all plaques in an assay independently of their diameter and shape), count and size measurements of viral plaques is laborious, tedious and time consuming. Moreover, the repetitive nature of this procedure can lead to an increase in error rate. In this sense, in the past decades, improved accessibility of scanner and digital cameras has facilitated the automation of the image analysis step, improving speed and objectivity (Choudhry, 2016). However, the majority of software available either focuses on viral quantification (e.g., UFP/ml determination) (Sullivan et al., 2012) or needs specific experimental conditions (e.g., fluorescence microscopy) (Yakimovich et al., 2015; Culley et al., 2016; Katzelnick et al., 2018). Apart from that, in recent years other software has been developed for acquisition of particle size statistics and quantification of particles from a broad range of assays such as apoptosis in cultured cells (Helmy & Azim, 2012), clonogenic assays (Cai et al., 2011) and counting of cell, bacterial and yeast colonies and tumor spheroid particles (Geissmann, 2013; Choudhry, 2016). Nonetheless, these programs have been designed for specific assays with different image conditions and are not fit to accurately characterize plaques of viral origin.

We developed ViralPlaque, a versatile ImageJ macro for automated detection and analysis of viral plaques. We demonstrate that this method is fast, accurate and suitable on images obtained from different sources such as a cell phone camera or a flatbed scanner. It is tunable in several parameters, like size of plaques and measurements to perform. Lastly, adaptation of ViralPlaque to the user’s particular experimental conditions is incorporated through a machine-learning plugin.

Material and Methods

Plaque assay

Plaque assays were performed on baby hamster kidney cells (BHK-21 clone 13; ATCC CCL10) and on African green monkey kidney cells (Vero, ATCC CCL81) as previously described (García Núñez et al., 2010). Cells were maintained at 37 °C and 5% CO2 in Dulbecco’s modified Eagle’s medium (DMEM, Life Technologies, Grand Island, NY, USA) supplemented with 10% fetal bovine serum (FBS) and antibiotics (Gibco-BRL/Invitrogen, Carlsbad, CA, USA). Viruses used for this study were foot-and-mouth disease virus (FMDV) isolates of serotype A and vesicular stomatitis Indiana virus (VSV). For FMDV plaque assays, virus dilutions (0.2 ml per well of a 6-well tissue culture plate) were added onto a cell monolayer containing 106 BHK-21 cells seeded the day before, and the plates were incubated for 1 h at 37 °C to allow virus internalization. Then, the virus inoculum was removed and the cells were overlaid with 2.5 ml of semisolid medium containing SeaPlaque Agarose 0.8% (Lonza, Rockland, ME, USA) and DMEM supplemented with 2% FBS. At 48 h postinfection (hpi), cells were fixed with 4% formaldehyde and stained with crystal violet. Plates were washed with tap water, dried and scanned using a flatbed office scanner at 150 or 1,200 dots per inch (dpi). For VSV plaque assays, virus dilutions (0.1 ml per well of a 24-well tissue culture plate) were added onto a cell monolayer containing 3 × 105 Vero cells seeded the day before, and the plates were incubated for 1 h at 37 °C to allow virus internalization. Then, the virus inoculum was removed and the cells were overlaid with 1 ml of semisolid medium containing methylcellulose 0.7% and DMEM supplemented with 2% FBS. At 72 hpi, cells were fixed with 4% formaldehyde and stained with crystal violet. Plates were washed with tap water, dried and placed over a white background (a white sheet of paper) with the back of the plate facing upwards. Plates were digitalized using a 13 Megapixels cell phone camera and natural lighting. Direct illumination was avoided (i.e., flash from cell phone was turned off). Alternatively, plates can be placed over a light box illuminator to improve the outline of the plaques.

Description of ViralPlaque

ViralPlaque is written in ImageJ Macro language (IJ), which is a scripting language that allows controlling many aspects of ImageJ (Schindelin et al., 2012). Programs written in this language can be used to perform a desired set of algorithms over the image, which include variables, control structures (conditional or looping statements) and user-defined functions. In addition, the IJ allows access to all available ImageJ functions and to a vast number of built-in functions. ViralPlaque is available for download at https://sourceforge.net/projects/viralplaque/.

An outline of ViralPlaque usage is illustrated in Fig. 1. Once the macro is installed, the basic workflow is to open an image file and then run ViralPlaque (Fig. 1A). It will ask for specific parameters, method and mode to run and then it performs a set of predefined steps over a user-defined area of the image being analyzed. The macro includes three methods of image analysis, namely LowRes, Difference, and Weka. The LowRes method was developed to work on low resolution images digitalized using a scanner set at low dpi or simply obtained with a cell phone camera. The other two methods (Difference and Weka) were designed specifically for high resolution images (1,200 dpi) obtained from a flatbed scanner. Regarding the running mode, ViralPlaque includes two modes (single well and 6-well). The former (recommended) requires the user to select the area of a culture plate to be analyzed. In case the plaque assay is performed in 6-well plates and the whole plate is digitalized, the user may select the 6-well mode to increase analysis throughput (only available for LowRes and Difference methods).

Figure 1 Basic workflow for ViralPlaque (A) and Viral Plaque-Batch (B) macros.

Blue filling color indicates that the step requires user input.

An alternative macro, the Viral Plaque-Batch macro (Fig. 1B) allows the user to reuse the functionality of the software on more than one image, increasing throughput even more. Once the batch macro is run, the user is asked to indicate both input and output directories. Then, the ViralPlaque macro is run sequentially on every image of the input directory (the user can change run parameters each time). Finally, once measurements are performed, a results file is automatically saved in the output directory for each image processed.

Specific instructions for installing and running both macros including step-by-step overview for each method and mode are listed in File S1.

Implementation of ViralPlaque

A step-by-step description of the methods included in ViralPlaque is depicted in Fig. 2. The LowRes method is the fastest one and includes a set of five major steps (Fig. 2A). Firstly, two filters are applied to the image, namely Median filter and Gaussian Blur filter. The user is prompted to choose the radius (in pixels) for each filter. Of note, the radius for the Median filter should not be larger than the diameter of the smallest plaque to be detected and the radius for the Gaussian Blur filter should be close to one quarter of the size of the Median radius. Default values were chosen from the images tested in this work; the user should set the values that most fit the input images. Next, thresholding is performed in order to convert the image into black and white. This step can be set to run automatically (default is manual) though this hinders precision. Then, several processes of denoising are performed (erode, dilation, and fill holes) previous to the segmentation step of the watershed command. Watershed is a widespread technique for image segmentation; this step allows the macro to identify as two different objects plaques that are in contact (i.e., merged). Finally, IJ command ‘Analyze Particles’ is run and the user can control the results obtained (i.e., plaques contours are displayed non-destructively on the duplicate image) manually before proceeding to execute the measure command. Video S1 exemplifies step-by-step usage of this method. Additionally, if there is no need to obtain size measurements, this method includes a Count Only option where only the number of plaques detected will be informed. This option is designed specifically to count plaques so some parameters will be overwritten to specific values (File S1).

Figure 2 Summary of the processing steps for the three alternative methods included in ViralPlaque macro.

(A) Low resolution; (B) high resolution. Filling color indicates the condition of each step: blue, manual or automated modes available (default is manual); green: optional step; orange, parameters can be set at the initial prompt. ROI, region of interest.

The Difference method is also a fast method that includes a set of six major steps (Fig. 2B). First, the ImageJ ‘Enhance Contrast’ function is run, followed by a sharpening process. Users can change some of these default parameters at the prompt window each time the macro is run. The next step is finding the edges over a duplicate of the image. Then, and image calculator is used to create a new image based on the difference between the duplicate and the original image (hence the name of the method). Next, as in the LowRes method, thresholding, denoising and ‘Analyze Particles’ steps are performed. Video S2 exemplifies step-by-step usage of this method.

Lastly, the Weka method consists of roughly similar steps as the Difference method though no manual thresholding is performed (Fig. 2B). To circumvent this step, the Trainable Weka Segmentation plugin (Arganda-Carreras et al., 2017) was used to train classifiers on example images in order to obtain a classifier file (File S1). Alternatively, the user can execute the Trainable Weka Segmentation plugin on its own example images in order to obtain a classifier file. For specific instructions, see File S1. Additionally, Video S4 exemplifies step-by-step procedure of this training process. Once the macro is run, it produces a duplicate image that will be classified using the Weka plugin. The image is then converted to black and white followed by denoising, filling holes and segmentation steps. Finally, ‘Analyze Particles’ command is run and manual control of the results obtained can be done. Video S3 exemplifies step-by-step usage of this method.

Results

We tested the ability of the ViralPlaque macro to detect and measure viral plaques accurately from low resolution images. To this end, digital images of wells (n = 18) recorded using a flatbed scanner (150 dpi) from FMDV plaque assays were used. Firstly, manual measurements of the area of individual plaques (n = 151) were performed with ImageJ’s ‘draw ellipse’ and ‘measure’ tools (Analyze-Measure) on the original images. Then, ViralPlaque macro was used to measure the area of individual plaques using the LowRes method with default parameters in single well mode (Fig. 3A). All false positive plaques (i.e., plaques assigned by ViralPlaque that did not represent actual lysis plaques) were eliminated manually before the measurement step using the ROI manager as described in File S1. In some cases, plaques were not detected automatically by the IJ macro (for example, see plaque G in Fig. 3A); in other cases, two adjacent plaques were erroneously considered as a single plaque (see plaque C in Fig. 3A and Fig. S1B). Nonetheless, 129 of the manually detected plaques were identified using the automated method, which gives a recall of 0.854. Moreover, the 10th and 90th percentile values of the plaque area proved to be close to those obtained for the manual measurements, suggesting good reliability (Fig. 3B and Fig. S1A). Indeed, a good linear correlation was observed between manual and automated analysis (R2 = 0.831), with a slope value of 1.17 as represented in Fig. 3C.

Figure 3 Comparison of plaque measurements with manual and LowRes methods.

(A–D) Analysis of 6-well FMDV plaque assays scanned using a flatbed office scanner at 150 dpi. (E–H) Analysis of 24-well VSV plaque assays digitalized using a 13 Megapixels cell phone camera. Median filter radius was increased to 8 px and Gaussian Blur filter radius was set at 2 px. The original low-resolution images (A, E) were processed with the ViralPlaque macro (B, F). The plaques detected by the macro are circled in yellow and labeled (B, F). In (B), arrows indicate plaques that were only detected manually (G and H). (C, G) Distribution of plaque areas. Box-plots represent data obtained between 10th and 90th percentiles; median area is indicated by a horizontal line. The number of plaques detected by each method is given in brackets. (D, H) Correlation between manual and digital measurements of individual plaque area. The equation of the linear least-squared fit and goodness of fit R2 is given for each method.

In order to test the versatility of ViralPlaque, we tested the macro on images of VSV plaque assays recorded with a cell phone camera. In this case, plaque assays were performed on 24-well culture plates and using a methylcellulose overlay. Manual measurements of the area of individual plaques (n = 152) were performed and compared to the dimensions of plaques detected by the LowRes method. As indicated in Material and Methods, in this case the radius for the Median and the Gaussian filters were increased in order to better suit the analysis to the image resolution (Fig. 3D). As shown in Fig. 3E, 134 of the manually detected plaques were identified using the automated software (recall = 0.881). Again, plaque area 10th and 90th percentile values were similar to those obtained for the manual measurements (Fig. 3E and Fig. S1D). As with the FMDV assay, a good linear correlation was observed between manual and automated analysis (R2 = 0.794), with a slope value of 1.018 (Fig. 3F).

Next, we tested the ViralPlaque macro on high resolution images using both Difference and Weka methods with default parameters in single well mode (Figs. 4A and 4B). Since these image conditions increase sensitivity thus allowing for more false positive results, minimum plaque size was increased to 300 px2. Most of the manually detected plaques were identified using the automated software, with recall values of 0.883 and 0.834 for Difference and Weka methods, respectively. Again, plaque area presented a similar distribution independently of the method used, suggesting good reliability (Figs. 4C and 4E). As expected, a very good linear correlation was observed between manual and automated analysis, as represented in Fig. 4D (slope = 0.990, R2 = 0.9399 for manual vs. Difference method; slope = 1.085, R2 = 0.9314 for manual vs. Weka method).

Figure 4 Comparison of Difference and Weka methods with manual measurements.

Analysis of 6-well FMDV plaque assays scanned using a flatbed office scanner at 1,200 dpi. High-resolution original image (A) and images processed by the manual method (B) and by IJ macro Difference (C) and Weka (D) methods. The plaques detected by the macro are circled in yellow and labeled from A to F. (E) Area of the six plaques detected in (A) as determined by Manual, Difference and Weka methods. (F) Distribution of plaque areas of the total number of plaques analyzed from 17 images. Box-plots represent data obtained between 10th and 90th percentiles; median area is indicated by a horizontal line. The number of plaques detected by each method is given in brackets. (G) Correlation between manual and digital measurements of the area of individual plaques. The equation of the linear least-squared fit and goodness of fit R2 is given for each method. (H) Distribution of plaque areas from 17 images measured manually and using ViralPlaque macro methods. Boxes represent distribution of data between 10th and 90th percentiles; horizontal lines indicate median values.

Remarkably, manual measurements of viral plaques took an average of 100 s per image, though this is highly dependent on the number of plaques per well. In turn, the average processing time per image using the macro was 30 s for the Weka method and even 15 s for LowRes and Difference methods, representing a >3-fold and 5-fold reduction, respectively. Together, the high reliability, robustness and speed of the ViralPlaque macro support its utility for the automated calculation of viral plaque dimensions.

Discussion

Imaging programs ideally should be flexible with tunable parameters defined by the user and cost free. Having this in mind, we chose Fiji as a platform, which is a distribution of the popular open-source Java-based image processing program ImageJ focused on biological image analysis (Schindelin et al., 2012). Fiji combines powerful software libraries with a broad range of scripting languages which, in turn, enables rapid implementation of image-processing algorithms and extensive plugins and macros for specific purposes. In this sense, ViralPlaque implements this powerful tool to facilitate a tedious task. Also, the macro includes a prompt stage where the majority of the relevant parameters (including circularity, particle size, enhance contrast, set measurement) can be changed by the user to better fit its needs. Also, if the scale of the image is known, the ratio of pixels per mm can be easily given at this stage, thus prompting the results to be informed in millimeters. Even so, if more specific changes were needed, the macro developed can be easily modified to fit the user’s specific requirements. In line with this, as part of the ImageJ software, Fiji has a large and hands-on user community that could aid improving ViralPlaque.

Admittedly, machine learning is a fast-growing method of data analysis that automates analytical model building. It is a branch of artificial intelligence based on the idea that systems can learn from data, identify patterns and make decisions with minimal human intervention. In this sense, we applied a previously developed plugin (Arganda-Carreras et al., 2017) to train classifiers that could sort pixels of the image between two classes: cells and background (lytic plaque). The classifier file obtained is called upon by the macro to accurately detect and measure the viral plaques. Moreover, the user has the potential to train their own classifiers, adjusting the macro for the specific conditions of the experiment at hand, thus severely improving performance. Nonetheless, it should be kept in mind that the Weka method is computationally expensive thus its use is highly dependent on the hardware capabilities at disposal. On the other hand, LowRes and Difference methods are much simpler algorithms (and thus much faster) making them more suitable for most every day computer hardware. It should be noted that those methods are, by default, run with auto-thresholding option off, this means that user input is needed to threshold each image (e.g., selecting the corresponding values of pixels that better separate cells from plaques). Alternatively, this option can be set on, hence reducing user intervention (more reproducibility) but heavily increasing false positive and false negative rates depending on particular image conditions (i.e., illumination, artifacts, contrast).

Additionally, a batch-mode macro is also presented. Once the user has reached the best parameters and method for the images to be processed, this macro is an alternative that should help increasing speed even further. This is achieved by eliminating repetitive steps such as saving results files and by automatically opening all image files in a particular folder.

Conclusions

ViralPlaque is a fast, open-source, accurate, tunable, and user-friendly image analysis method for the obtention of viral plaques statistics that highly replicates manual measuring and facilitates characterization of cytolytic viruses.

Supplemental Information

Figure S1 Analysis of FMDV (A, B) and VSV (C, D) plaque area of individual culture wells

(A, D) Boxes represent distribution of data between 10th and 90th percentile; horizontal lines indicate median values. (B, C) Comparison of manual and automated measurement of plaque area for representative lysis plaques. In (B), plaque C (area = 1,142 px2 as detected by ViralPlaque) actually represents two lysis plaques (C and H) with areas of 782 and 366 px2 that were erroneously recorded as a single plaque. Plaques G and H were not detected by the IJ macro.

Click here for additional data file.

File S1 Instructions for downloading and installing ViralPlaque

Click here for additional data file.

Video S1 Step-by-step usage of LowRes method

Click here for additional data file.

Video S2 Step-by-step usage of Difference method

Click here for additional data file.

Vidoe S3 Step-by-step usage of Weka method

Click here for additional data file.

Video S4 Steps for training new classifiers from user data (Trainable Weka Segmentation)

Click here for additional data file.

M.C. is a fellow of the ANPCyT and A.C. is a doctoral fellow of the National Research Council (CONICET) at the University of Luján, Argentina. M.I.G. is member of CONICET Research Career Program. We thank Sandra Cordo, María Cruz Miraglia and Sabrina Amalfi for kindly providing images and Matías Richetta for helpful discussions.

Additional Information and Declarations

Competing Interests

Author Contributions

Data Availability

The authors declare there are no competing interests.

Marco Cacciabue conceived and designed the experiments, performed the experiments, analyzed the data, prepared figures and/or tables, authored or reviewed drafts of the paper, approved the final draft.

Anabella Currá conceived and designed the experiments, prepared figures and/or tables, authored or reviewed drafts of the paper, approved the final draft.

Maria I. Gismondi analyzed the data, contributed reagents/materials/analysis tools, authored or reviewed drafts of the paper, approved the final draft.

The following information was supplied regarding data availability: The raw data is available at Sourceforge: https://sourceforge.net/projects/viralplaque/.

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
