# Peer review of "ViralPlaque: a Fiji macro for automated assessment of viral plaque statistics"

_PeerJ, doi:10.7717/peerj.7729_

## Round 0.1 · original submission · Major Revisions

The reviewers did find value in the approach but do bring up a number of points/concerns such as repeatability and details of the methods that should be addressed in the revision.

Reviewer 1 ·

Basic reporting

Plaque assay is a classic method for counting viral infectious particles while manual counting of viral plaques is a time-consuming task. The authors aim to develop a software system ViralPlaque to automatically recognize and analyze plaque pictures obtained from a flatbed scanner or a cell phone camera. A machine learning plugin has also been integrated in the analysis algorithm, which is considered to be an application of artificial intelligence (AI).

Experimental design

1) In aspect of biological methods, what’s the success rate of your plaque assay method? As far as we know, traditional plaque assays are not easy to conduct and highly depend on the experiences of operators.
2) How to address the problem of plaque fusion from the aspect of image recognition? You know, sometimes two or more plaques can usually merge to form a larger plaque in disks or plates.
3) I don’t know how you take pictures by a cell phone. Was the light source from the back of 6-well plates or from the same side of the cell phone? If the light source is from the back of 6-well plates, the outline of plaques would be clearer than those the light source from reflected light. A flatbed scanner relies on a reflected light source, to our knowledge.

Validity of the findings

The algorithm used in image analysis may be reasonable. The more important things are the repeatability of plaque assays, as well as how you get the raw data of the images. More details regarding these two parts should be described in your manuscript.

Additional comments

The paper is well written and gives an insight into application of technology of digital image recognition, which is similar to face recognition or AI doctor in analyzing CT films. But the latter is based on huge demands while yours is not. For example, there are hundreds of CT film need radiologists to read every day and an AI doctor in image recognition will help finish most of their work. To emphasize the importance of your research, other demands of plaque assay should be mentioned, not just viral titer testing or plaque morphology and dimensions regarding the replication kinetics and the virulence of a particular virus. Especially, when these demands need huge image data, and aim to solve a practical problem.

Reviewer 2 ·

Basic reporting

The plaque assay has been a mainstay of virology research for >50 years and, for some viruses, it remains the best way to measure infectivity. In this generally well written article, Cacciabue et. al. present a new semi-automated image analysis approach for the quantification of viral plaques. Written as a plugin for the commonly used image analysis software Fiji (ImageJ), the method will likely be of use to virology researchers.

Experimental design

The figures outline the design of ViralPlaque and validate it by comparison to traditional manual quantification methods. In this way, the authors provide a reasonable demonstration of ViralPlaque’s ability to make reliable measurements. However, there is focus on quantifying the size of plaques; a reliable completely automated method of counting the number of plaques would also be of use to researchers.

Validity of the findings

I was able to download and run the plugin and some example data provided on the author’s source forge site. Having watched the supplementary videos it was easy to start taking measurements. I also downloaded an assortment of images of different plaque assays from google images and was able to perform some reasonable quantification of these. Therefore, I am satisfied that the approach is working as the authors suggest.

Additional comments

1. The approach seemed to be very dependent on users manually thresholding the image. This limits its ability to perform automated image analysis. If the macro could also perform some kind of background subtraction to create a homogenous background this may improve the approach.

2. I’m not sure sufficient detail is provided to allow users to use the WEKA method - maybe some notes on training the WEKA segmentation on different data?

3. Whilst running the macro multiple times I seemed to accumulate objects in the ROI manager, the authors might want to include a line of code to clear the ROI manager between runs?

4. Figure 2 multiple incorrect spellings of ‘conversion’.

External reviews were received for this submission. These reviews were used by the Editor when they made their decision, and can be downloaded below.

---

## Round 0.2 · accepted · Accept

A reviewer pointed out that recent improvements in methods have led to the creation of larger data sets that would benefit from the approach you are developing here and suggests that you may want to mention such benefits in the discussion. This is merely a suggestion and you are welcome to incorporate this into your manuscript if you so chose, though it is by no means required.

Reviewer 1 ·

Basic reporting

As image recognition techniques, this research really make sense.

Experimental design

No comments

Validity of the findings

No comments

Additional comments

The authors have responded to almost all of my concerns. They added that they used the classical plaque method firstly described by Dulbecco in 1952. This classical method has been developed for many years since its repeatability is not very high due to its complicated operation. In some experiments, overlay media have been changed into methyl cellulose or Avicel. Yin et al have used the improved method to produce viral plaques in 96-well plates (PMID: 25515071). They also further developed this improved method to screen out viral inhibitors from a compound library (PMID: 30900754). Their ultimate goal is to develop a high throughput antiviral drug system based on plaque inhibitory tests. Candidate inhibitors will also undergo concentration gradient plaque experiments. These attempts will generate large amounts of image data, which really need your ViralPlaque macro. Therefore,in your manuscript, a highly repeatable method as well as huge data production will substantiate the importance of your research. At least, these method developments and their new uses should be mentioned in your discussion section.